# Spatial localization ability of planarians identified through a light maze paradigm

Renzhi Qian[1]◉, Yuan Yan[1]◉, Yu Pei[1]◉, Yixuan Zhang[1], Yuanwei Chi[1], Yuxuan Chen[1], Kun Hao[1], Zhen Xu[1], Guang Yang[1], Zilun Shao[1], Yuhao Wang[1], Xinran Li[1], Chenxu Lu[1], Xuan Zhang[1], Kehan Chen[2], Wenqiang Zhang[2], Baoqing Wang[1], Zhengxin Ying[1], Kaiyuan Huang[1]*

1 College of Biological Science, China Agricultural University, Beijing, China, 2 College of Engineering, China Agricultural University, Beijing, China

◉ These authors contributed equally to this work.
* cfhuangkaiyuan@gmail.com

**Data Availability Statement:** All relevant data are within the paper and its Supporting information files.

**Funding:** The author(s) received no specific funding for this work.

## Abstract

Spatial localization ability is crucial for free-living animals to fit the environment. As shown by previous studies, planarians can be conditioned to discriminate directions. However, due to their simplicity and primitiveness, they had never been considered to have true spatial localization ability to retrieve locations of objects and places in the environment. Here, we introduce a light maze training paradigm to demonstrate that a planarian worm can navigate to a former recognized place from the start point, even if the worm is transferred into a newly produced maze. This finding identifies the spatial localization ability of planarians for the first time, which provides clues for the evolution of spatial learning. Since the planarians have a primitive brain with simple structures, this paradigm can also provide a simplified model for a detailed investigation of spatial learning.

## Introduction

Spatial localization, the ability to learn and remember locations and routes, is crucial to survival for most animals. The ability can be used to forage for food, return to sites of storage or safety, and avoid sites of danger, thus helping animals survive and fit the environment. Thus this ability is considered to have evolved in almost all species [1].

Planarians are free-living flatworms that usually live and stick under rocks, debris and water plants in streams, ponds, and springs [2]. They are considered one of the first class of animals to have a centralized brain structure, and they had evolved various sensory abilities, including sensitivity to light [3], temperature [4], water currents [5], chemical gradients [6], vibration [7], magnetic fields [8] and electric fields [9]. For freely living in the environment, it is reasonable to hypothesize that planarians have evolved spatial localization ability. The study of planarian behavior mainly focused on simple classical conditioning and proved that planarians can learn simple associative tasks. Moreover, planarians are concluded to have abilities to discriminate directions in a Y or T maze [10]. With such ability, they can turn their moving directions in response to a certain stimulus. However, due to the primitiveness and simplicity

**Competing interests:** The authors have declared that no competing interests exist.

of planarians, they are not considered to have the ability of true, complex spatial localization ability [11], which is used to localize a certain place in a familiar environment.

To prove the hypothesis that planarians have evolved spatial localization ability, we designed a light maze paradigm inspired by the Morris water maze paradigm [12]. The Morris water maze, designed by Morris in 1981, is widely used to evaluate the rats' spatial localization ability through distal localization (when the goal object is invisible, inaudible and cannot be detected by smell) rather than proximal localization (when the goal object is visible, audible or detectable by smell) [12]. The Morris water maze uses water as the aversive stimuli to prompt the rats to swim to a platform. Since planarians have innate behavior to avoid light [3], similar to the Morris water maze, we chose light as the aversive stimuli to prompt planarians to move to the dark chamber in the maze. The planarians' primitive eyes can only sense the light rather than imaging; thus, they cannot visualize the dark chamber. To avoid any chemical cues that a worm might leave in the dark chamber, in the test phase, worms were transferred into newly manufactured mazes from their original mazes. Thus, the information that a worm use to navigate to the dark chamber can only be distal. In the light maze, the distal cues provided to the worm could be such as the topography of the maze, the light intensity gradient, or the geomagnetic field which the goal object did not provide direct information to the worm. The proximal cues could be such as chemicals from the goal object which can be directly smelled by the worms. In the text, we use the word "escape" to show that the worms solved the maze and successfully entered the dark chamber. We also use the escape latency to measure the worms' performance, which is most commonly used in the Morris water maze, defined as the time for the animal to find the dark chamber and escape the maze. Our light maze paradigm is also inspired by the Barnes maze, which uses open spaces as aversion stimuli to prompt rodents to seek shelter.

Here we demonstrated that planarians were able to use distal information to localize a dark chamber in the light maze. To evaluate the learning effect of the worms, we measured the escape latencies and drew typical routes of worms which showed learning effects. These are the first findings suggesting that planarian worms have spatial localization abilities, indicating that planarians might have higher cognition ability that had never been considered. As a primitive forerunner of more evolved animals, the study of planarian cognition can be evolutionarily instructive to the study of more evolved animals. Also, since the planarians' nervous system has simple structures, it can provide a model for spatial learning studies.

## Materials and methods

### Planarians

A laboratory strain of *D. japonica*, originating from wild collected *D. japonica* (identified by cytochrome c oxidase subunit 1 gene) from the Cherry-Valley in Beijing Botanical Garden, Haidian district, Beijing, China in 2019. Worms were maintained in Montjuic Water(Simulated components of fresh water from Montjuic) in the dark and fed with chicken liver twice a week on Monday night and Thursday night. The procedure in planarian rearing used protocol from M. Shane Merryman et al. 2018 for reference [13]. The length of planarians used in the experiment varied from 1 cm to 1.5 cm.

### Experimental setup

To investigate the spatial localization ability of the planarians, we designed a light maze. It is a square maze with a dark chamber near one corner (Fig 1A). The maze consists of a square pool with a side length of 65 mm and a depth of 6mm, the corners of the maze are rounded to make it easier for planarians to crawl. A square dark room with inner side length of 14 mm

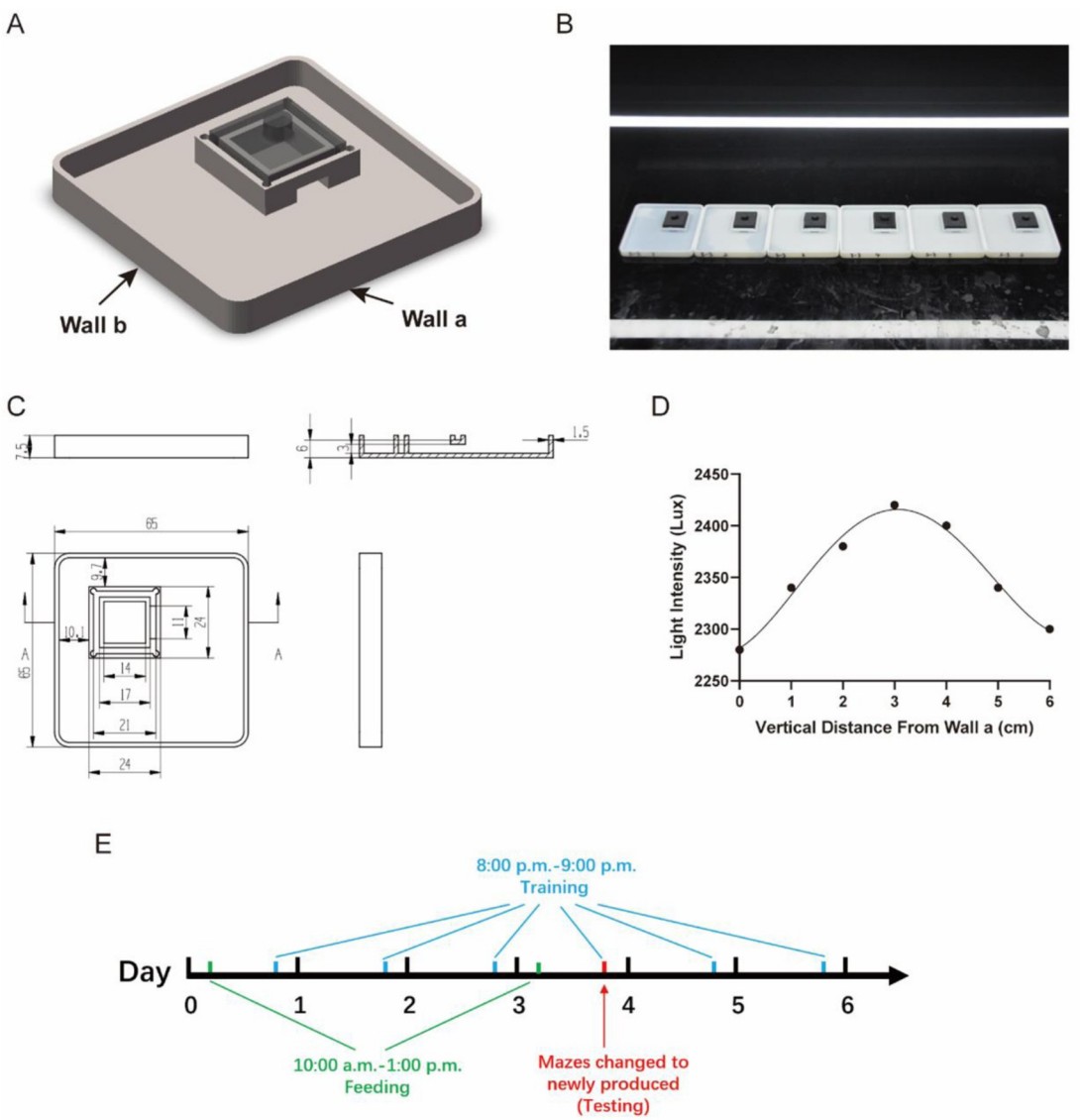

**Fig 1. Experimental setup and training procedure.** A. The conceptual drawing of the light maze. B. The experimental setup of the light maze paradigm. C. The blueprint of the light maze. The upper right shows the lateral view of section A. Length unit: mm. D. Light intensity distribution of the maze. The light intensity distribution is measured according to vertical distance from Wall a. Light intensity variance along Wall a is less than 5 Lux, which can be neglected. E. Diagram of the training procedure.

and outer side length of 24 mm is located in one of the quadrants as the destination of the maze. The starting position is located in the corner of the opposite quadrant of the dark chamber. The dark chamber has only one entrance. 15 mL Montjuic water is added to the maze before starting a trial.

The mazes were 3D-printed. The material used for 3D printing is photosensitive resin and PLA (Polylactic acid), the main body of the mazes was printed in white using photosensitive resin, and the dark chamber lid was printed in black using PLA (The dark chamber lid can be detached from the main body of maze). In the SLA printing method, a photocurable and photosensitive polymeric material is placed on a surface apparatus and is subsequently opened to a UV light to cure the resin to form a initial layer. When initial layers cure, the repetition of this

process fabricates the 3D printing product [9]. The toxicity released by the stereolithography (STL) printing materials (photosensitive resin in this experiment) is fatal to worms, which causes the worms to die and disintegrate in hours, even in the mazes handled by ultraviolet to have reduced toxicity. Although the fused deposition modeling (FDM) materials like PLA is non-toxic, it is of lower printing precision, and its water leaking problem may cause worms to escape from the maze while training. We found that a biocompatible encapsulation material called Parylene to totally block the toxicity from the STL printed parts, using which can make worms safely live in the maze without health problems for over 2 weeks. The inner side of the mazes was coated with Parylene film (thickness of 12 μm) to block the toxicity from the resin [14]. The structure and parameters of the maze are shown in (Fig 1C).

During training, the mazes were put in a line on a black acrylic board to avoid light reflections, and each maze was filled with 15 mL of Montjuic Water. A LED lighting tube was set above the entrance of the dark chamber (Fig 1B) to make the entrance lighted and avoid direct guidance to the entrance by the light. The length of the LED tube was 1.2 m, and the power of the LED tube was 13 W. The distance between the bottom of the LED lighting tube to the acrylic board was 125 mm. The LED lighting tube was set right above the dark chamber. Light intensity distribution of the maze is shown in Fig 1D. Worms were carefully manipulated through a smooth woolen brush.

## Training procedure

We totally used 36 worms (2 worms were excluded due to self-fission in the training process) and they were divided into 6 groups. We used 6 mazes to train each group of worms in the first 3 trials. In trial 4, all mazes used to train any of the worms were changed to newly manufactured (36 new mazes). The training procedure started at 8:00 p.m. and ended at 9:00 p.m. each day for a total of 6 trials for each worm in 6 days. 1 day before the training, worms were taken out from the home well to a petri dish. Diagram of the training procedure is shown in Fig 1E. To control worms' satiety, they were fed at 10:00 a.m. on day 1 and day 4 in a 12-well plate, each worm in a single well. Before training, worms were taken to a 12-well plate, then the water in the mazes is changed and the mazes were set up as described in the last section. The worm is put 5 mm away from wall a and wall b (the starting point might be a little different in each trial because the worm need to sink to the bottom). The room temperature during training was controlled at 20±1˚C. After the training, the worms were not taken out, and the maze along with the acrylic board was put in an incubator to control the environment temperature.

## Statistical analysis

36 worms were involved in training and 2 of them were excluded due to self-fission. However, possibly because the worms were gradually getting habituated to the light and the environment, about one third of the worms in each trial quickly got to rest and stop moving in each trial except trial 1. Hence, our data excluded worms that rest for more than 20 mins in 40 mins in each trial. So, from trial 1 to trial 6, n = 34, 24, 24, 24, 23, 22. The latency of other worms which did not enter the dark chamber (either rested or kept on moving) were counted as 40 mins. Also, worms that went into the dark chamber but got out within 30 mins did not count as a successful escape.

For statistical analysis, because the escape latencies in trial 1 and trial 4 were not normally distributed (Kolmogorov–Smirnov test), we used the nonparametric Mann–Whitney U-test to evaluate statistical significance of latencies in trial 1 and trial 4. One way ANOVA was applied to determine whether there is a statistical significance between escape latencies of trial 2, 3, 4, 5

and 6. One way ANOVA and subsequent Turkey's multiple comparison test was applied to determine whether worm speed in trial one is significantly higher than other trials. All data were analyzed using PRISM (GraphPad Prism 9.0.0(121)). Routes are drawn by hand with Photoleap (2.13).

### Ethics statement

The overall study and animal experiments of this manuscript conformed to the guidelines and regulatory standards of the Institutional Animal Care and Use Committee of China Agricultural University and approved by the Institutional Animal Care and Use Committee of China Agricultural University.

## Results

We tested for recall of the location of the dark chamber of the worms in trial 4 when the mazes were changed to newly manufactured. In trial 4, because the mazes were new, worms cannot use local cues such as chemicals for navigation. The latency result shows that worms in trial 4 displayed a significantly shorter time to escape compared with trial 1 (two-tailed U-test, P<0.01; Fig 2A). To test whether this behavior is stable after changing the maze, we continued trial 5 and trial 6. Comparing escape latencies in trial 4 to trial 2, 3, 5 and 6, there is no significant difference, which means that changing to newly manufactured maze did not affect worm's performance in the maze, showing that the worms might not use chemical cues in this case (One way ANOVA, P>0.1; Fig 2A). We also counted the worms that underwent successful escapes. From trial 3 to trial 6, there were 10–12 worms that successfully escaped the maze (Fig 2C). However, from trial 2 to trial 6, a worm might successfully escape the maze in some trials but not in other trials. From trial 2 to trial 6, the number of worms that successfully escaped 0, 1, 2, 3, 4 and 5 times is 3, 2, 17, 8, 3 and 3, and the number of worms that consecutively could not escape the maze for 2, 3, 4, and 5 times is 12, 8, 1 and 3.

To exclude the effect of worm speed on latency, we counted the worm speed of a total moving distance of 241 mm (the perimeter of the maze) for worms which moved for more than 241 mm, otherwise, we used its actual moving distance to count its speed. The result shows that worm speed in trial 1 are significantly higher than other trials (One way ANOVA, P<0.001; Fig 2B), which means the worm speed did not shorten the latencies from trial 2 to trial 6.

To further understand the worms' learning effect, we showed representative routes of trial 1 and trial 4 of 2 worms which successfully escaped the maze in trial 4 (Fig 2C–2F). In trial 4, the mazes are newly manufactured, and no chemicals or mucus can affect the navigation of the worms. As the worms did not know the location of the dark chamber prior to the training, the routes of trial 1 were mainly distributed on the walls of the maze. In trial 4, the worms that showed learning effects quickly secede from the wall of the maze and find the entrance of the dark chamber, which showed significantly shorter routes than routes in trial 1, demonstrating that worms can use distal information to navigate to their destinations.

## Discussion

In our findings, we designed a light maze to demonstrate that planarians were able to use distal information to localize a dark chamber in the light maze. After training, the worms can use shorter time or shorter distances to navigate to the dark chamber. Below, I'll discuss details and issues in this paradigm.

From the 1950s to the 1960s, numerous studies had been done to understand planarian learning and behavior [15]. The most commonly used procedure is a classical conditioning

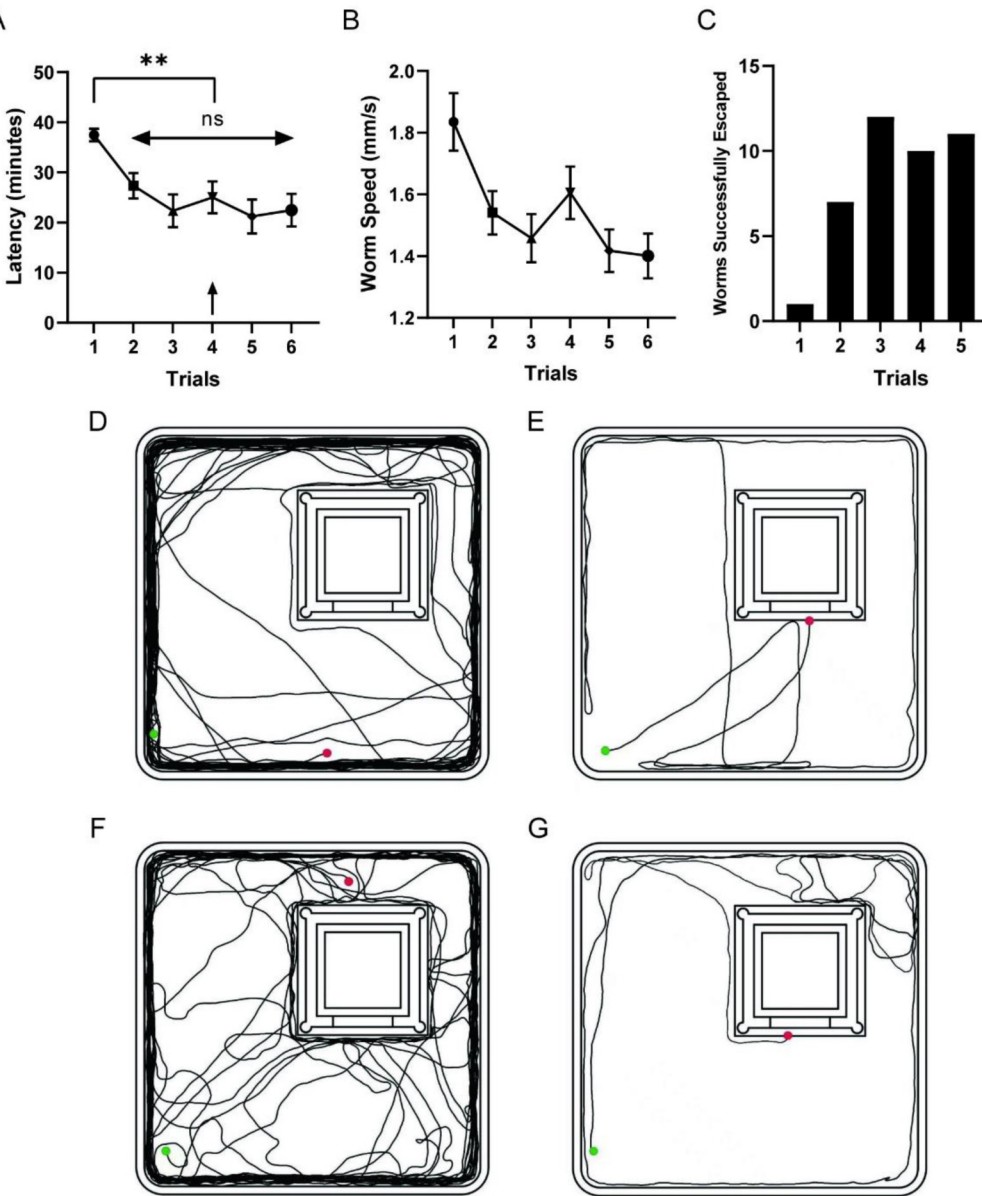

**Fig 2. Results of the light maze experiment.** A. Latency of each trial. Data are the average±SEM of worm's latency for worms moved for more than 20 mins in 40 mins. From trial 1 to 6, n = 34, 24, 24, 24, 23, 22. B. Worm speed in each trial. C. Worms successfully escaped in each trial. C-G. Representative routes of learned worms. C, E show routes of trial 1. D, F show routes of trial 4. C and D are routes of 1 worm and E, F are routes of 1 worm. The green spots show the start point and the red spots show the end point.

protocol to make worms associate a light stimulus with an electrical stimulus [16]. Other studies showed that worms can learn more complicated tasks such as discriminating directions in a Y or T maze [10, 17]. However, due to the manual practical difficulties and the control of variables, a large number of research results failed to be reproduced in lots of cases [15]. Consequently, this field became largely abandoned, and even the ability of planarians to form long-term memory was questioned [18].

Some classical conditioning paradigms of the planarians can be attributed to pseudoconditioning or sensitization instead of true learning and memory encoded by the brain [19]. Since spatial learning is a more complex behavior that requires the integration of different neural circuits in higher animals [20], spatial learning might also be a great point to investigate the memory and learning of planarians, and can be regarded as true learning and memory.

The control of variables and manipulations of the experimenters can also largely affect the worms' behavior. In previous studies, light intensity [21], water temperature [22], water existence [10], time of day [22], time of year [23], chemical components of water [23], chemical components of food [24], worm's appetite level [25], slime trails [26], worm fatigue state [23], magnetic fields [27], training conditions and manipulation of the experimenter [23] were considered to affect the worms' behavior. Our light maze paradigm was designed to decrease the number of variables as much as possible to make the experiment easy to reproduce. In our paradigm, the light intensity, water components, water temperature, appetite level and training time of day are strictly controlled.

The fabrication of mazes using 3D printing techniques is a great advantage in this research. Designing such a number of mazes of different shapes requests a much higher cost than 3D printing techniques. With the assistance of the 3D printing techniques, further investigation of the worm's spatial learning will be much more convenient.

The results demonstrated that worms showed learning effects in each trial. However, a worm might successfully escape the maze in some trials but not in other trials, which makes it a little hard to judge the learning effect of the worm. We believe that this phenomenon might be caused by multiple factors including the habituation phenomenon. I When the worms habituate to the environment and light stimulation, it might ignore the light and choose to rest under the light. Or this is because the navigation ability of planarians might be primitive, and cannot accurately solve every task. So, many worms cannot solve the task in some trials.

Although we endeavored to exclude proximal cues that worms might use to navigate, we still do not understand which kind of distal cues the worms might have utilized. We speculate that the worms used the light intensity gradient combined with the geometric information of the maze to navigate to the dark chamber. The worms might learn to get to brighter place to find the dark chamber. Also, the worms can learn to secede the wall to find the dark chamber. To further investigate this question, we think that the position of the dark chamber can be changed to test whether learned worms can still get into the dark chamber. Also, the electric and magnetic gradients from the led, earth and/or other lab equipment could also be used as distal cues for worms to navigate.

In conclusion, we identified the spatial localization ability of planarians by presenting a paradigm using the light maze we designed. The paradigm can also be a new tool for the analysis of spatial learning in planarians. In this work, planarian worms were found to use distal cues instead of proximal cues to navigate to the dark chamber. Since planarians are now the most primitive animal found to have spatial learning ability, identifying the spatial localization ability of them might provide insights into the evolution of spatial learning.

## Supporting information

**S1 Data.**
(XLSX)

## Author Contributions

**Conceptualization:** Renzhi Qian, Yuan Yan, Kaiyuan Huang.

**Data curation:** Renzhi Qian, Yuan Yan, Yu Pei, Yixuan Zhang, Guang Yang, Zilun Shao, Yuhao Wang, Xinran Li, Chenxu Lu.

**Formal analysis:** Renzhi Qian, Yuan Yan, Yu Pei, Guang Yang, Zilun Shao, Yuhao Wang, Xinran Li, Chenxu Lu, Xuan Zhang.

**Investigation:** Renzhi Qian, Yuan Yan, Yixuan Zhang, Yuanwei Chi, Yuxuan Chen, Kun Hao, Zhen Xu.

**Methodology:** Renzhi Qian, Yuan Yan, Yixuan Zhang, Yuanwei Chi, Yuxuan Chen, Kun Hao, Zhen Xu, Kehan Chen, Wenqiang Zhang, Kaiyuan Huang.

**Project administration:** Kaiyuan Huang.

**Resources:** Kehan Chen, Wenqiang Zhang.

**Supervision:** Kaiyuan Huang.

**Writing – original draft:** Kaiyuan Huang.

**Writing – review & editing:** Baoqing Wang, Zhengxin Ying.

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
