## [Decision Letter · Decision Letter 0]

22 Dec 2022

PONE-D-22-32791Spatial Localization Ability of Planarians Identified Through a Light Maze ParadigmPLOS ONE

Dear Dr. Huang,

Thank you for submitting your manuscript to PLOS ONE. After careful consideration, we feel that it has merit but does not fully meet PLOS ONE’s publication criteria as it currently stands. Therefore, we invite you to submit a revised version of the manuscript that addresses the points raised during the review process.

Dear Dr Huang,

Thank you very much for your submission of Spatial Localization Ability of Planarians Identified Through a Light Maze Paradigm to PLOS ONE. Both reviewers have carefully read and commented on your manuscript, I agree with their comments and thus recommend for Major Revisions. Importantly, as highlighted by both reviewers the statistical minutia needs to be developed further and be fully and clearly reported. Similarly the plots ought to be made more clear. At this stage in the submission process, all comments highlighted by both reviewers are valid and ought to be addressed in order for this manuscript to fully meet the requirement for publication at PLOS ONE. 

Kind regards,

Dr Elias Garcia-Pelegrin

PLOS Academic Editor

We look forward to receiving your revised manuscript.

Kind regards,

Elias Garcia-Pelegrin, Ph.D

Academic Editor

PLOS ONE

Journal Requirements:

4. Please upload a copy of Figure 2G, to which you refer in your text on page 7. If the figure is no longer to be included as part of the submission please remove all reference to it within the text.

Reviewers' comments:

Reviewer's Responses to Questions

**Comments to the Author**

1. Is the manuscript technically sound, and do the data support the conclusions?

Reviewer #1: Yes

Reviewer #2: No

2. Has the statistical analysis been performed appropriately and rigorously? 

Reviewer #1: No

Reviewer #2: No

3. Have the authors made all data underlying the findings in their manuscript fully available?

Reviewer #1: Yes

Reviewer #2: No

4. Is the manuscript presented in an intelligible fashion and written in standard English?

Reviewer #1: Yes

Reviewer #2: Yes

5. Review Comments to the Author

Reviewer #1: In this paper, the authors develop a light maze and investigate the spatial localization ability of planarians. They found that planarians succeeded in navigating themselves to a learnt position, suggesting that planarians could learn and remember locations and routes. This paper can be published in a modified form in the future. However, I cannot recommend publication in its present state. Before the paper is considered for publication, there are questions that must be answered:

Major concerns:

Is it correct that successful worms follow the same trajectory as the previous successful trials? Why not calculating the trace accuracy using the data? In fact, you show only two figures regarding the data analysis. Maybe too little analysis? Linked to this, is there a difference regarding the speed of worms between the first trial and the other trials? I wonder the speed of worms is affected by the learning accuracy.

I could not find any statistical explanations regarding the data in Results section. Linked to this, why not adding some statistical analysis to results shown in fig 2B? Are there any statistical differences regarding the ratio of worms reached criterion between trial 1 and the other trials?

I wonder what kind of spatial information worms rely on. What happens if some geometry arrangement in the light maze is modulated after several trials? Also, you may as well consider the rearrangement of the position of the dark chamber after several trials. Could you expand discussion regarding this?

Minor comments:

Training procedures: The training procedures is also written in the first paragraph of Results section. I recommend you that you should move it to Method section. In addition, I found that you said in the “Training Procedure” of Method section that 24 worms were used in the training. However, in Result section, you said that 35 worms were totally used. Which is correct?

Fig1B: Can you make fig1B a little bit larger? In addition, the two boxes on the far left appear to be different in color from the others. Why?

fig2C-F: Although you present some trajectories, I would like to see some movies.

Reviewer #2: The paper suggests a new light-maze system for studying spatial orientation in Planarians.

The idea and the system are interesting and has merits. The paper itself however have some serious flows. The first and most important is that the entire statistics section is completely missing.

There is indeed a part titled "statistics", but it contains no relevant information whatsoever. The results parts follows the same theme – there is a paragraph titled "results" but in actuality it is just an extension of the "Methods" part. Also, because the paper contains no statistical tests, there are no real results reported anywhere in it.

These two issues alone should disqualify the paper, to my opinion.

However, because of the somewhat novel and interesting system – I think it is worth publishing after some major revisions. Mainly: (a) adding the full statistics, (b) rewriting the "Results" and "Methods", and (c) measuring light levels on the maze surface (there is no need to re-run the entire experiment – only arrange the set up as it was and measure light levels across the maze, maybe at the beginning, middle and just above the dark solution patch).

There are also various other issues in the paper which I will list below.

Introduction.

Line 15: Please elaborate more or at least give an example regarding the differences between the "ability to discriminate directions" and "complex localization ability".

Line 15-24: This part deals with the Morris water maze and the light-based system the authors offer. The authors' suggestion and design are solid, but they are in fact a combination of the Morris water maze and the Barnes light maze. I therefore feel like the latter should be at least mentioned here.

Line 24-25: ("The planarians’ primitive eyes can only sense the light rather than imaging; thus, they cannot visualize the dark chamber". I am not sure 'visualize' is the correct word here and would have chosen a slightly different phrasing.

Line 26: The text jumps to introduce "trial 4" without describing first the experimental design or mentioning previous trials or their purpose. I would (a) consider moving this entire paragraph to "Methods" (b) change "trail 4" to "test phase" or something similar along these lines.

Line 31: "This finding proves that the planarian worms have spatial localization ability, which is not yet discovered". One cannot claim to report something which is not yet discovered. I would rather write something like "these are the first finding suggesting that planarian worms have spatial localization abilities".

Lines 34-35: ("Also, since the planarians’ nervous system has simple structures, it can

provide a model for spatial learning studies."). This sentence is true and it makes sense. However, it remains at the statement level if there is no follow up, elaboration or a suggestion for further research. Similar statements repeats through the paper, but still without any concrete suggestions.

Materials and Methods.

Line 3-4: First, past form is preferable here (worms were instead of worms are). Second, I feel like a short explanation of what exactly "Montjuic Water" are is due here. I for once don't know this term and reference 13 is locked behind a pay-wall. Also, if this is indeed a standard procedure in planarian rearing I would like to see some references to this method: for example "Worms were maintained in Montjuic Water in the dark (like in XX et-al. 2019, YY & ZZ 2022) etc…."

Finally, food amounts and preparation methods are missing: How much chicken liver? In what form? Was it whole? Crushed? Minced? Was it mixed in the water? Was it removed the next day?

Experimental Setup.

Line 2: Why are the maze dimensions not mentioned in the text? Please describe the maze and mention them before you are referring me to the Figures (although they should be mentioned there too)

Line 3: Extra 'r' in 'were' (here the paragraph is back to past tense, which is good). Also, why was photosensitive resin used for the maze? It is probably nothing and I am sure the authors have a good explanation and they thought this out. But since this paper deal with light mazes and it effect on small creatures I would have like to see this issue somehow been addressed. It would be reassuring to the reader to explain why it is not a problem.

Line 6: ("The inner side of the mazes is coated…"). We are back to present tense, please don't jump from tense to tense, at least not within the same paragraph.

Line 10-15: This discusses the lightning of the maze. In light maze experiments it is common to measure and report the illumination level (in Lux), rather than state the specifications of the lightning apparatus. If specifications already appear, I would also like to see the makers/brand of the Led tube reported – This could help a lot in recreating the experiment.

In light of the the results mentioned in the Discussion - It would be a good idea to re-set the experiment (no animals or water needed) and to measure and report the light level at least at the beginning and end points of each maze. Otherwise the authors cannot claim the worms relied only on distal cues to solve the maze.

Training procedure:

This entire part is very unclear and inconsistent. I feel it should be re-written completely and therefore there are no line numbers stated here but only general remarks.

Some unclear and troubling points:

1. this sections starts with: "We used 6 mazes to train 24 worms in the first 3 trials".

Were the worms trained individually or in groups of 4? Did water or mazes were switched between training runs? (Something like that was mentioned before - but it is not mentioned here).

2. Next the text states: "total of 6 trials in 6 days" - isn't this come up to a total of 34 trials (6 treatments X 4 trials equals 24 trials). What is the reason for this discrepancy?

3. "To control worms’ satiety, they were fed at 10:00 a.m. on day 1 and day 4"

When were the worm fed? On Sunday and Thursday as mentioned before? A day before training? Were Sunday and Thursday both were a day before training? This is very unclear.

Also - Why is important to control satiety when there is no food reward in the maze and the initiative to solving it is light avoidance? If it is only for standardization, say so. If there is another reason - mention it.

4. "Then the water in the mazes is changed (add to 15 mL) and the mazes were set up as described above." – Well, (a) nothing is described above, (b) were the water changed or were water added to fill a certain minimum? Also, why?

5. While re-writing this section, it might be useful to add an explaining diagram describing trials, times, number of worms and the entire experiment process.

Statistical Analysis:

This is the entire statistical analysis section: "Statistical Analysis All data were analyzed using PRISM (GraphPad Prism 9.0.0(121)). Routes are drawn by hand with Photoleap (2.13)."

Need I say more? Please add Statistics…

Also, please explain the process of drawing the routes, as I understand it - Photoleap is a drawing software, not a movement tracking one.

Results:

This part should also be re-written. In reality it is not a result section but a continuation of the "Methods" part. This is probably somewhat due to the fact that since the statistical tests are missing, there is nothing to report.

This part should contain only the results of each important step of the experiments (including the test-names, the correct statistic for each test, the number of repetitions/iterations and the numbers of animals used in each such tests).

This entire sections should be re-written and moved to the methods, so here too – I will address some issues in general. Some of these are important and alarming.

1. The numbers of worms and trials here are different from what is described earlier (in the methods section).

Why were 35 worms here and either 24 or 34 in previous part?

The executions times are also different from what is mentioned in the methods (although this might be the result of an am/pm typo). If this is the results section – how come a different number of worms was used? If this is actually a continuation of the Methods section – why is it under Results?

2. Worms' satiety is mentioned again here – the question remains, why is it important in the framework of a light maze?

3. "During the training process, only **partial worms** showed learning and got to the dark chamber **fast** in each trial." (a) I assume "partial worms" means "some worms"? Please rephrase. (b) What does fast mean here? Is it important? (explain why), is it trivial? (omit it).

Is "fast" on a scale? Is it only binary fail/success? More importantly - are Fast/Slow definitions meaningful within your experimental design? If so, they need to be defined, measured and reported accordingly.

Lines 3-4: "In each training trial, a planarian was put in the start point (opposite the dark chamber corner)". Please add this start point to the figures. Do this not only in the hand drawn route figures (in which the starting point slightly differs - but also in the relevant places in Figure 1.

Line 14: "If a worm stopped moving outside the dark chamber for more than 20 mins, its data from that single trial is excluded" – What does that mean? I think it means that worms that didn't move for more than 20 min. were omitted from the statistics. But I can't be sure. Also, because you don't report the statistics, it is hard to understand if or how meaningful it is. Please explain/rephrase and include all relevant information (including the number of cases where this happened). Also elaborate about your "learning criterion" while at is. When were 40min. used and where did you use 20 min.?

Line 16-17: "From trial 3 to trial 6, there were 10-12 worms that showed learning effects in each trial (Fig2.B). This part starts to resemble "results". Please phrase it more clearly. Also you have to mention (clearly) somewhere what is your exact "learning criterion" (Is it "worm got to the shade in under 20min.? something else? Don't leave me guessing) and add it to the graph/figure. Also add the statistical test results (is 7 in trial 2really statistically different from 10 in trial 4? Should I guess here too?).

Lines 17-18: "Comparing trial 4 to trial 2, 3, 5 and 6, there is no significant difference, showing that the worms might not use chemical cues in this case (Fig2.A)". Here you are jumping to a different graph showing something different than the one two lines ago, yet you are still speaking about trial numbers only (rather than training/test runs or other more informative terms).

This is very confusing and should be corrected. More important, you are claiming there are significant differences here but do not bother to back this statement up: What test did you use? What were the results? Where are the statistical values?

Discussion:

Lines 3-4: Why are these lines relevant here? It has nothing to do with either the lines preceding or following it.

Lines 6-8: examples needed here - also, are you sure this belongs in the discussion and not in the introduction?

Lines 10-14: You are comparing between planarians and mice here. While It is true that one animal have a simpler brain than the other, just stating this fact does not make it a "great investigation point" – please back this statement up by either explaining why is it such a great opportunity and/or at least suggest a theoretical way to test this.

Lines 14-16: the lines state "As in our paradigm, we applied the light maze paradigm to identify the spatial localization ability of the planarians, which can be regarded as true learning and memory."

First, the words "As in our paradigm" does not connect to the previous sentence. Second, please explain what do you mean by "true learning and memory" (are there false learning and memory within your paradigm? Do you mean to differ it from other kind of learning? If so – which? And why?). Otherwise just omit this part.

Line 24-25: "The results demonstrated that…"

1. Normally this is how you should have started the discussion.

Here however there are no results. What appears under "Results" is actually a continuation of the methods section and the Statistics and Results sections are completely missing.

2. "A worm might show learning effects in some trials but not in other trials." – While this can happen, you can't just wave this aside without addressing the issue. You need to state why do you think it happened, how often, what does it mean, and if it is a problem or not. Although some explanations follow, they are confused, not sufficient and does not address some of the issues I just mentioned. For example, you cannot just say "the planarians navigation system is primitive and that is why they can solve some tasks and not others". Please explain at least what tasks you are referring to (not to mention what sets apart the unsolved ones). Also, from the initial phrasing it follows that worms also differed in learning results within the same task. Please rephrase to address these issues.

Lines 34-35: The authors suggests the "worms used the light intensity gradient combined with the geometric information of the maze to navigate to the dark chamber." – This is another reason why it is important to measure the light gradient across the experiment arena. As already mentioned above – this is something I like to see done before submitting this paper again. Also, how exactly was the geometric of the maze used by the worms?

Finally, as was mentioned by the authors earlier in the article – the worms can detect also vibration, and magnetic and electric fields, how did they account for, or controlled these variables? (i.e - no talking in the experiment chamber, isolating electric devices etc...)

Line 36: "we identified the spatial localization ability of planarians by presenting a new paradigm using the light maze we designed." – No you did not, you cannot make this claim without presenting the statistics supporting it. Also, as mentioned above - this paradigm is far from being new. If anything, you applied an existing paradigm to a new model animal.

Lines 38-39: "In this work, planarian worms were found to use distal cues instead of proximal cues to navigate to the dark chamber. I am sorry, but I have to ask again – can you really say that? The authors themselves just stated light gradient as a possible explanation for their findings, and a few other factors (electric and magnetic gradients from the led, earth and/or other lab equipment) could also have influence the results. Please measure the light gradient and also rephrase this part.

Lines 40-41: "Identifying the spatial localization ability of such a primitive invertebrate might provide insights into the evolution of spatial learning". Once again, this stays at the statement level. Even if this is true, please elaborate how this provide insights, what kind of insights do you think it will provide, what do you suggest as a direction for future research, where do you think this should go next – things like that..

Figures: Figure 2C,D,E,F – These suppose to represent the learned routes of the worms. However, as the text indicates they were drawn by hand using Photoleap. Since Photoleap is not a movement tracking software but rather a 'freehand' drawing program, I would like to see more information regarding the methods by-which the worms movement was translated from the test arena to the final figures. Also, why is the start point in each of them slightly different?

6. PLOS authors have the option to publish the peer review history of their article (what does this mean?). If published, this will include your full peer review and any attached files.

Reviewer #1: No

Reviewer #2: No

---

## [Author Response · Author response to Decision Letter 0]

31 Jan 2023

Major concerns:

Is it correct that successful worms follow the same trajectory as the previous successful trials? Why not calculating the trace accuracy using the data? In fact, you show only two figures regarding the data analysis. Maybe too little analysis? Linked to this, is there a difference regarding the speed of worms between the first trial and the other trials? I wonder the speed of worms is affected by the learning accuracy.

We thank the reviewer for pointing out these issues. However, after trying some software, we could not find an effective tracing software that can accurately trace the worms. Because when the worm climbs on the sidewall of the maze, the software loses the target. This might also happen when a worm wriggle and causes its body to change shape. Therefore, we chose to draw some routes to prove that the worms could somehow get to the dark chamber. Also, by roughly estimating the worm’s trace, we did not find a general rule for how the worms got to the dark chamber. Even so, our video can be provided and used for analysis by other researchers. We also counted the speed of the worms and found that worms are getting slower after the first trial, which leads to a respectively longer latency for worm to get to the dark chamber. Data analysis of the speed of the worms is now added in the result part. 

I could not find any statistical explanations regarding the data in Results section. Linked to this, why not adding some statistical analysis to results shown in fig 2B? Are there any statistical differences regarding the ratio of worms reached criterion between trial 1 and the other trials?

We thank the reviewer for pointing out this issue. We apologize for omitting statistical analysis. Statistical analysis of the original data is now added to the result part.

I wonder what kind of spatial information worms rely on. What happens if some geometry arrangement in the light maze is modulated after several trials? Also, you may as well consider the rearrangement of the position of the dark chamber after several trials. Could you expand discussion regarding this?

We thank the reviewer for this suggestion. Due to technical restrictions on our route tracing method, we are still working on this question. The discussion part is expanded regarding to the suggestion.

Minor comments:

Training procedures: The training procedures is also written in the first paragraph of Results section. I recommend you that you should move it to Method section. In addition, I found that you said in the “Training Procedure” of Method section that 24 worms were used in the training. However, in Result section, you said that 35 worms were totally used. Which is correct?

We thank the reviewer for pointing out this issue. The number of worms is now corrected.

Fig1B: Can you make fig1B a little bit larger? In addition, the two boxes on the far left appear to be different in color from the others. Why?

We thank the reviewer for pointing out this issue. Fig1B is now larger. The photosensitive resin material tends to change its color to a little yellow when exposed under light. However, since planarians do not have visual abilities, we don’t think that this factor might affect the result.

fig2C-F: Although you present some trajectories, I would like to see some movies.

We thank the reviewer for this suggestion. The movie can be provided.

Reviewer #2: The paper suggests a new light-maze system for studying spatial orientation in Planarians.

The idea and the system are interesting and has merits. The paper itself however have some serious flows. The first and most important is that the entire statistics section is completely missing.

There is indeed a part titled "statistics", but it contains no relevant information whatsoever. The results parts follows the same theme – there is a paragraph titled "results" but in actuality it is just an extension of the "Methods" part. Also, because the paper contains no statistical tests, there are no real results reported anywhere in it.

These two issues alone should disqualify the paper, to my opinion.

However, because of the somewhat novel and interesting system – I think it is worth publishing after some major revisions. Mainly: (a) adding the full statistics, (b) rewriting the "Results" and "Methods", and (c) measuring light levels on the maze surface (there is no need to re-run the entire experiment – only arrange the set up as it was and measure light levels across the maze, maybe at the beginning, middle and just above the dark solution patch).

There are also various other issues in the paper which I will list below.

We thank the reviewer for pointing out these issues. The statistics were added and the relevant parts were rewritten as suggested.

Introduction.

Line 15: Please elaborate more or at least give an example regarding the differences between the “ability to discriminate directions” and “complex localization ability”.

We thank the reviewer for the suggestion. The relevant part is now elaborated more as suggested.

Line 15-24: This part deals with the Morris water maze and the light-based system the authors offer. The authors’ suggestion and design are solid, but they are in fact a combination of the Morris water maze and the Barnes light maze. I therefore feel like the latter should be at least mentioned here. 

We thank the reviewer for the suggestion. The Barnes light maze is now mentioned in the text.

Line 24-25: (“The planarians’ primitive eyes can only sense the light rather than imaging; thus, they cannot visualize the dark chamber”. I am not sure ‘visualize’ is the correct word here and would have chosen a slightly different phrasing.

We thank the reviewer for pointing out this issue. The sentence is now rephrased as “The dark chamber is not visible to the worms.”

Line 26: The text jumps to introduce “trial 4” without describing first the experimental design or mentioning previous trials or their purpose. I would (a) consider moving this entire paragraph to “Methods” (b) change “trail 4” to “test phase” or something similar along these lines.

We thank the reviewer for pointing out this issue. The phrase “trail 4” is changed to “test phase”.

Line 31: “This finding proves that the planarian worms have spatial localization ability, which is not yet discovered”. One cannot claim to report something which is not yet discovered. I would rather write something like “these are the first finding suggesting that planarian worms have spatial localization abilities”.

We thank the reviewer for pointing out this issue. The sentence is rephrased as suggested.

Lines 34-35: (“Also, since the planarians’ nervous system has simple structures, it can

provide a model for spatial learning studies.”). This sentence is true and it makes sense. However, it remains at the statement level if there is no follow up, elaboration or a suggestion for further research. Similar statements repeats through the paper, but still without any concrete suggestions.

We agree. In fact, currently, there is little research about planarian behavior, nor did we find any neural circuit studies of planarians, possibly due to the difficulty of gene editing in planarians. Therefore, we could not currently give concrete suggestions and we hope that further studies can solve this problem to help establish this field. 

Materials and Methods.

Line 3-4: First, past form is preferable here (worms were instead of worms are). Second, I feel like a short explanation of what exactly “Montjuic Water” are is due here. I for once don’t know this term and reference 13 is locked behind a pay-wall. Also, if this is indeed a standard procedure in planarian rearing I would like to see some references to this method: for example “Worms were maintained in Montjuic Water in the dark (like in XX et-al. 2019, YY & ZZ 2022) etc….”

Finally, food amounts and preparation methods are missing: How much chicken liver? In what form? Was it whole? Crushed? Minced? Was it mixed in the water? Was it removed the next day?

We thank the reviewer for pointing out this issue. We actually used protocol from M. Shane Merryman et al. 2018 for reference and this case is now illustrated in this part.

Experimental Setup.

Line 2: Why are the maze dimensions not mentioned in the text? Please describe the maze and mention them before you are referring me to the Figures (although they should be mentioned there too)

We thank the reviewer for pointing out this issue. The maze dimensions are now added to the text.

Line 3: Extra ‘r’ in ‘were’ (here the paragraph is back to past tense, which is good). Also, why was photosensitive resin used for the maze? It is probably nothing and I am sure the authors have a good explanation and they thought this out. But since this paper deal with light mazes and it effect on small creatures I would have like to see this issue somehow been addressed. It would be reassuring to the reader to explain why it is not a problem.

We thank the reviewer for pointing out this issue. The explanation of this issue is now added to the discussion part.

Line 6: (“The inner side of the mazes is coated…”). We are back to present tense, please don’t jump from tense to tense, at least not within the same paragraph.

We thank the reviewer for pointing out this issue. The tense is now corrected.

Line 10-15: This discusses the lightning of the maze. In light maze experiments it is common to measure and report the illumination level (in Lux), rather than state the specifications of the lightning apparatus. If specifications already appear, I would also like to see the makers/brand of the Led tube reported – This could help a lot in recreating the experiment.

In light of the the results mentioned in the Discussion – It would be a good idea to re-set the experiment (no animals or water needed) and to measure and report the light level at least at the beginning and end points of each maze. Otherwise the authors cannot claim the worms relied only on distal cues to solve the maze.

We thank the reviewer for pointing out this issue. The lighting conditions is now added in the method part.

Training procedure:

This entire part is very unclear and inconsistent. I feel it should be re-written completely and therefore there are no line numbers stated here but only general remarks.

Some unclear and troubling points:

We thank the reviewer for pointing out the issues below. The entire part is now rewritten and clearly illustrated the issues below.

3. this sections starts with: “We used 6 mazes to train 24 worms in the first 3 trials”.

Were the worms trained individually or in groups of 4? Did water or mazes were switched between training runs? (Something like that was mentioned before – but it is not mentioned here).

2. Next the text states: “total of 6 trials in 6 days” – isn’t this come up to a total of 34 trials (6 treatments X 4 trials equals 24 trials). What is the reason for this discrepancy?

3. “To control worms’ satiety, they were fed at 10:00 a.m. on day 1 and day 4”

When were the worm fed? On Sunday and Thursday as mentioned before? A day before training? Were Sunday and Thursday both were a day before training? This is very unclear.

Also – Why is important to control satiety when there is no food reward in the maze and the initiative to solving it is light avoidance? If it is only for standardization, say so. If there is another reason – mention it.

Satiety control is for standardization and is illustrated in the discussion part.

4. “Then the water in the mazes is changed (add to 15 mL) and the mazes were set up as described above.” – Well, (a) nothing is described above, (b) were the water changed or were water added to fill a certain minimum? Also, why?

Water is changed. Now, the water amount added to the maze is illustrated in the Experimental Setup part.

5. While re-writing this section, it might be useful to add an explaining diagram describing trials, times, number of worms and the entire experiment process.

A diagram is now added.

Statistical Analysis:

This is the entire statistical analysis section: “Statistical Analysis All data were analyzed using PRISM (GraphPad Prism 9.0.0(121)). Routes are drawn by hand with Photoleap (2.13).”

Need I say more? Please add Statistics…

Also, please explain the process of drawing the routes, as I understand it – Photoleap is a drawing software, not a movement tracking one.

We thank the reviewer for pointing out this issue. We apologize for omitting statistical analysis. Statistical analysis of the original data is now added to the result part.

Results:

This part should also be re-written. In reality it is not a result section but a continuation of the “Methods” part. This is probably somewhat due to the fact that since the statistical tests are missing, there is nothing to report.

This part should contain only the results of each important step of the experiments (including the test-names, the correct statistic for each test, the number of repetitions/iterations and the numbers of animals used in each such tests).

This entire sections should be re-written and moved to the methods, so here too – I will address some issues in general. Some of these are important and alarming.

We thank the reviewer for pointing out these issues. The entire section is now rewritten, and problems described below were all solved.

3. The numbers of worms and trials here are different from what is described earlier (in the methods section).

Why were 35 worms here and either 24 or 34 in previous part?

The executions times are also different from what is mentioned in the methods (although this might be the result of an am/pm typo). If this is the results section – how come a different number of worms was used? If this is actually a continuation of the Methods section – why is it under Results?

The number of worms is now corrected. The entire section is now rewritten.

2. Worms’ satiety is mentioned again here – the question remains, why is it important in the framework of a light maze?

Satiety control is for standardization and is illustrated in the discussion part.

3. “During the training process, only **partial worms** showed learning and got to the dark chamber **fast** in each trial.” (a) I assume “partial worms” means “some worms”? Please rephrase. (b) What does fast mean here? Is it important? (explain why), is it trivial? (omit it).

Is “fast” on a scale? Is it only binary fail/success? More importantly – are Fast/Slow definitions meaningful within your experimental design? If so, they need to be defined, measured and reported accordingly.

The entire part is now rewritten and the sentences relevant were rephrased. 

Lines 3-4: “In each training trial, a planarian was put in the start point (opposite the dark chamber corner)”. Please add this start point to the figures. Do this not only in the hand drawn route figures (in which the starting point slightly differs – but also in the relevant places in Figure 1.

The start point is now illustrated in the text.

Line 14: “If a worm stopped moving outside the dark chamber for more than 20 mins, its data from that single trial is excluded” – What does that mean? I think it means that worms that didn’t move for more than 20 min. were omitted from the statistics. But I can’t be sure. Also, because you don’t report the statistics, it is hard to understand if or how meaningful it is. Please explain/rephrase and include all relevant information (including the number of cases where this happened). Also elaborate about your “learning criterion” while at is. When were 40min. used and where did you use 20 min.?

The relevant part is now rephrased and clearly illustrated.

Line 16-17: “From trial 3 to trial 6, there were 10-12 worms that showed learning effects in each trial (Fig2.B). This part starts to resemble “results”. Please phrase it more clearly. Also you have to mention (clearly) somewhere what is your exact “learning criterion” (Is it “worm got to the shade in under 20min.? something else? Don’t leave me guessing) and add it to the graph/figure. Also add the statistical test results (is 7 in trial 2really statistically different from 10 in trial 4? Should I guess here too?).

The relevant part is now rephrased and clearly illustrated.

Lines 17-18: “Comparing trial 4 to trial 2, 3, 5 and 6, there is no significant difference, showing that the worms might not use chemical cues in this case (Fig2.A)”. Here you are jumping to a different graph showing something different than the one two lines ago, yet you are still speaking about trial numbers only (rather than training/test runs or other more informative terms).

This is very confusing and should be corrected. More important, you are claiming there are significant differences here but do not bother to back this statement up: What test did you use? What were the results? Where are the statistical values?

The relevant part is now rephrased and clearly illustrated.

Discussion:

Lines 3-4: Why are these lines relevant here? It has nothing to do with either the lines preceding or following it.

We thank the reviewer for pointing out this issue. The sentence redundant and is now deleted.

Lines 6-8: examples needed here - also, are you sure this belongs in the discussion and not in the introduction?

We thank the reviewer for the suggestion. However, this point is too complicated to explain in this section and is well illustrated in the reference 15. Thus, we decide not to give an example here. And this part is relevant to the next paragraph that we are going to discuss, so we think that this belongs in the discussion part.

Lines 10-14: You are comparing between planarians and mice here. While It is true that one animal have a simpler brain than the other, just stating this fact does not make it a "great investigation point" – please back this statement up by either explaining why is it such a great opportunity and/or at least suggest a theoretical way to test this.

We thank the reviewer for the suggestion. However, there is currently zero investigation on behavior and neural circuit of the planarian worm. Thus, could not offer a theoretical way to test this. Anyway, we rephrased the sentences to make it sound reasonable.

Lines 14-16: the lines state "As in our paradigm, we applied the light maze paradigm to identify the spatial localization ability of the planarians, which can be regarded as true learning and memory."

First, the words "As in our paradigm" does not connect to the previous sentence. Second, please explain what do you mean by "true learning and memory" (are there false learning and memory within your paradigm? Do you mean to differ it from other kind of learning? If so – which? And why?). Otherwise just omit this part.

We thank the reviewer for pointing out this issue. The sentence is now rephrased.

Line 24-25: "The results demonstrated that…"

1. Normally this is how you should have started the discussion.

Here however there are no results. What appears under "Results" is actually a continuation of the methods section and the Statistics and Results sections are completely missing.

2. "A worm might show learning effects in some trials but not in other trials." – While this can happen, you can't just wave this aside without addressing the issue. You need to state why do you think it happened, how often, what does it mean, and if it is a problem or not. Although some explanations follow, they are confused, not sufficient and does not address some of the issues I just mentioned. For example, you cannot just say "the planarians navigation system is primitive and that is why they can solve some tasks and not others". Please explain at least what tasks you are referring to (not to mention what sets apart the unsolved ones). Also, from the initial phrasing it follows that worms also differed in learning results within the same task. Please rephrase to address these issues.

We thank the reviewer for pointing out these issues. The sentences are now rephrased.

Lines 34-35: The authors suggests the "worms used the light intensity gradient combined with the geometric information of the maze to navigate to the dark chamber." – This is another reason why it is important to measure the light gradient across the experiment arena. As already mentioned above – this is something I like to see done before submitting this paper again. Also, how exactly was the geometric of the maze used by the worms?

Finally, as was mentioned by the authors earlier in the article – the worms can detect also vibration, and magnetic and electric fields, how did they account for, or controlled these variables? (i.e - no talking in the experiment chamber, isolating electric devices etc...)

Line 36: "we identified the spatial localization ability of planarians by presenting a new paradigm using the light maze we designed." – No you did not, you cannot make this claim without presenting the statistics supporting it. Also, as mentioned above - this paradigm is far from being new. If anything, you applied an existing paradigm to a new model animal.

We thank the reviewer for pointing out this issue. The word “new” is now deleted.

Lines 38-39: "In this work, planarian worms were found to use distal cues instead of proximal cues to navigate to the dark chamber. I am sorry, but I have to ask again – can you really say that? The authors themselves just stated light gradient as a possible explanation for their findings, and a few other factors (electric and magnetic gradients from the led, earth and/or other lab equipment) could also have influence the results. Please measure the light gradient and also rephrase this part.

We thank the reviewer for pointing out this issue. The part is rephrased.

Lines 40-41: "Identifying the spatial localization ability of such a primitive invertebrate might provide insights into the evolution of spatial learning". Once again, this stays at the statement level. Even if this is true, please elaborate how this provide insights, what kind of insights do you think it will provide, what do you suggest as a direction for future research, where do you think this should go next – things like that..

We thank the reviewer for the suggestion. However, there is currently zero investigation on behavior and neural circuit of the planarian worm. Thus, we feel sorry that we could not expand this topic.

Figures: Figure 2C,D,E,F – These suppose to represent the learned routes of the worms. However, as the text indicates they were drawn by hand using Photoleap. Since Photoleap is not a movement tracking software but rather a 'freehand' drawing program, I would like to see more information regarding the methods by-which the worms movement was translated from the test arena to the final figures. Also, why is the start point in each of them slightly different?

We thank the reviewer for pointing out these issues. However, after trying some software, we could not find an effective tracing software that can accurately trace the worms. Because when the worm climbs on the sidewall of the maze, the software loses the target. This might also happen when a worm wriggle and causes its body to change shape. Therefore, we chose to draw some routes to prove that the worms could somehow get to the dark chamber. So, all the routes were drawn freehand through watching the video with Photoleap. The start point in each of them is slightly different because because the worm need to sink to the bottom, and this is when we started drawing.

---

## [Decision Letter · Decision Letter 1]

29 Mar 2023

PONE-D-22-32791R1Spatial Localization Ability of Planarians Identified Through a Light Maze ParadigmPLOS ONE

Dear Dr. Huang,

Thank you for submitting your manuscript to PLOS ONE. After careful consideration, we feel that it has merit but does not fully meet PLOS ONE’s publication criteria as it currently stands. Therefore, we invite you to submit a revised version of the manuscript that addresses the points raised during the review process.

Thank you for your revised version, please address and reply to the reviewers comments and concerns for this round in order for this manuscript to be reconsidered for publication.

We look forward to receiving your revised manuscript.

Kind regards,

Elias Garcia-Pelegrin, Ph.D

Academic Editor

PLOS ONE

Reviewers' comments:

Reviewer's Responses to Questions

**Comments to the Author**

1. If the authors have adequately addressed your comments raised in a previous round of review and you feel that this manuscript is now acceptable for publication, you may indicate that here to bypass the “Comments to the Author” section, enter your conflict of interest statement in the “Confidential to Editor” section, and submit your "Accept" recommendation.

Reviewer #1: All comments have been addressed

Reviewer #2: (No Response)

2. Is the manuscript technically sound, and do the data support the conclusions?

Reviewer #1: Yes

Reviewer #2: No

3. Has the statistical analysis been performed appropriately and rigorously? 

Reviewer #1: Yes

Reviewer #2: No

4. Have the authors made all data underlying the findings in their manuscript fully available?

Reviewer #1: Yes

Reviewer #2: No

5. Is the manuscript presented in an intelligible fashion and written in standard English?

Reviewer #1: Yes

Reviewer #2: No

6. Review Comments to the Author

Reviewer #1: The authors have satisfactorily addressed most of my concerns.

I wonder why worm speed in trial 1 are significantly higher than other trials.

You may or may not discuss this.

Reviewer #2: Lines 17-21: These lines suggests claims that the worms can navigate to a formerly recognized place from a start point. For these claims to hold water they need to meet at least one of two conditions: (1) did routes became shorter between runs? (There is no evidence in the text for that), (2) did solution times became shorter between runs (according to the authors answer to the first reviewer, it seems they actually got longer). Unless at least one of this conditions is satisfied, how do we know that the worms do not just randomly swim until they find the save dark patch?

Lines 50-51: I would rephrase these lines or completely omit them. (I think I can guess what the general meaning here is, but it is badly written and confusing and I am not sure how much it contributes to the overall understanding of the design).

Lines 84-85: Here I would have liked to see some reference to the photosensitive resin used. I asked for this matter (why use a photosensitive resin in such an experiment design when photosensitivity is key) to be clarified during the previous revision – and I am asking for this again now.

Line 107: please add the missing space between "changed' and "and".

Lines 117-119: The authors state "Light intensity variance along Wall a is less than 5 Lux, which can be neglected" - Reference is certainly needed here. Also light level should be measured not only from the wall of each maze but also in points between the maze 'entry' and maze 'solution'. From figure 1D it seems quite likely that there is might be some brightness gradient there.

Line 123: This is the first time the term "latency" appears in the text. I do not understand what it means here (or for that matter, when you use it again and again in the following parts of the text). Please define this terms clearly as it seems important to your claims. I would have preferred for it to be discusses or defined in the materials and methods section or in experiment design – but it doesn't really matter as long as you define it clearly the first time it appears in the text.

General remarks for the "statistics" chapter:

This is poorly written and does encourage trust in your results. Please phrase your research questions (preferably at the end of the introduction part), then report the statistical tests you chose to test these questions and why. This part should also include the number of repetitions and of worms in each state (not only of the ones used in the tests, but also the worms disqualified and the reasons for doing so).

Important: Note that you are mentioning specific tests in the "statistics" section and then report the result of *different* tests in the result section.

Lines 142-143: This would be a good place to state the total number (N) of the worm which the authors actually included in the statistics. Also, I do not completely understand what "latency' mean here too – please add an explanation. While we are on this subject – please state clearly not only the N for participating worms, but also for the disqualified ones.

Lines 145-145, 150-151: I would choose a term other than 'escape' and clearly define it. If you choose to remain with 'escape', please add a definition or an explanation for it (does it mean 'solved the light maze?', 'escaped the arena'? something else?). Also fix the extra dot/period in line 150. Note that in line 207 in the discussion (for example) you are using 'escape' in yet another context.

Lines 147-157:

a. these lines states: "To test whether this behavior is stable after changing the maze, we continued trial 5 and trial 6. comparing trial 4 to trial 2, 3, 5 and 6, there is no significant difference, showing that the worms might not use chemical cues in this case (One way ANOVA, P>0.1; Fig.2A)". The text does not specify what was compared to what?! What was the comparison criteria? Speed? Number of solutions? What is behind the statement that this comparison shows "that the worms might not use chemical cues in this case"

b. If there is significant difference between trial 1 and all other trails, indicate it - as you did in 2A. But this is a minor problem. If solving speed declines with trials (as stated in lines 154-157), how can you tell the worms learned? How can you say that there were more "solves" (if that is what 'escapes' means) in advanced trials because of learning and because of random chance (e.g., worms explored the maze for longer times).

c. I ask again, what exactly is "latency" in your experiment system? Also, why use 241mm. in some cases and actual moving distance in others? You are pooling two different data sets together and then running a single test on the new pool? Why?, Please explain this.

Lines 162-163: Please state how many worms we are talking about (this is already the results section and still we don't know how many worms were used for the statistics). What are the learning effects? As mentioned before – if solution times does not diminish and/or solution routes does not shorten, what is the criteria for learning?

Line 169: Please add a space between "40" and "min.

Lines 171-174: please fix this entire paragraph: the worms' routes Illustrations starts with 2D and not with 2C as appears in the text. More important issue here is that for some reason, in the advanced trials illustrations the end point of the worms *are not* under the dark chamber (while ending under it in the first trial). It seems like, at least from the illustrations that in opposition to the authors claims – the worms did not learn.

General remarks for the "Results" chapter:

I would like to see a better written Results part. This should include better definitions of what was measured. (for example: what is 'latency'? what are 'escapes'? how was 'speed' calculated? Why there is no report regarding routes length). Please describe exactly what was measured or tested under each such definition or terms. Do so before or when you are introducing this term for the first time.

You should also state what exactly was compared to what? Did solution times changed between trial 1 and 4? If yes - To what direction? If not – why and what criteria was compared? Why did you choose to compare trial 1 to 4 and not 1 to 6?

I would also like to see the actual number of animals eventually used in each statistical test as this is very unclear in the current text (something is mentioned under fig.2 but it is unclear). The N should appear also in the manuscript main text next to each relevant test. This is especially true when some of the N values mentioned are a bit too small for comfort).

Eventually, I cannot understand why you are describing statistical tests in the 'statistics' section and report results from completely different tests in the results. Please take care of this issue.

Lines 177-184: These lines belongs in the introduction. Please start the discussion with shortly describing your main findings. Then you can move on to explain them in relation to other work in the field or if there is none – their novelty. Either way though - these lines are not relevant here.

Line 192: It seems like the word 'including' is unnecessary here.

Lines 200-211: these lines also do not belong in the discussion. The process of manufacturing 3d mazes, their toxicity and the material used should appear in the introduction and in the method parts. You can of course stress the novelty and advantages of your design and techniques in the discussion – but not to delve into a deep discussion of materials and method. It belongs elsewhere. Also, please notice that in line 207, you are using 'escape' in yet another context.

Lines 212-214: These are the lines the discussion part should have begun with. However:

a. you are using 'escaped' in yet another context here.

b. It is not clear if the results are neither significant or actually support the authors claims.

c. This bring us back to the results section. Please rewrite it.

Lines 214-218: What does this mean for the results? How many times did this happen? This should be stated in the results section. Also, what do the authors mean by "worms cannot accurately solve every task"? Did they control for each worm in successive trials and singled out ones that repeatedly could not solve their light maze? Something else? Since there is no report of the proportions of 'solvers' vs. 'non-solvers' the reader can not evaluate the meaning or strength of this statement.

Lines 230-234: In these lines the authors claims that "In this work, planarian worms were found to use distal cues instead of proximal" and that "planarians are now the most primitive animal found to have spatial learning ability".

In light of the remarks above, I am yet to be convinced: What were the definitions for proximal and distal in this experimental setup? According to what hypothesis or tests the authors reached this results? These should appear in the manuscripts, ideally at the end of the introduction.

General remarks for the discussion:

The discussion also need some rewriting: The entire opening paragraph (lines 177-184) describes the history of the research field and belongs (if at all), in the introduction. The discussion then moves to explain reasons for the results (which should have been briefly summarized in the opening of the discussion). Directly after that the text explains the benefits (and some of the problems) of 3D printing and only on line 212 (35 lines into the discussion) finally some of the results first appear. This is structurally wrong and needs to be fixed.

What worries me more, however, is that what described in the 'results' section not necessarily shows that the worms actually learned between sessions: what is the criteria? Learning speed? Total numbers of worms that found the shelter in each trial? Shortening routes? Did the researchers followed specific worms and saw better performance? The results chapter lack clarity and the researches hypothesis are not mentioned, therefore it is hard to understand what on what bases the authors' states these results.

7. PLOS authors have the option to publish the peer review history of their article (what does this mean?). If published, this will include your full peer review and any attached files.

Reviewer #1: No

Reviewer #2: No

---

## [Author Response · Author response to Decision Letter 1]

1 May 2023

Reviewer #1: The authors have satisfactorily addressed most of my concerns.

I wonder why worm speeds in trial 1 are significantly higher than other trials.

You may or may not discuss this.

We thank the reviewer for pointing out this issue. When worms encounter an unfamiliar environment, their speed becomes high in order to get familiar with the environment as soon as possible, as worm speed in trial 1 is significantly higher than worm speed in other trial, and worms’ speed gradually decreases trials after trials. Interestingly, when worms are transferred from an old maze to a new maze (maze pattern is the same), their speed slightly increases, as worms’ speed in trial 4 are slightly higher than trial 2, 3, 5 and 6.

Reviewer #2: Lines 17-21: These lines suggest claims that the worms can navigate to a formerly recognized place from a start point. For these claims to hold water they need to meet at least one of two conditions: (1) did routes became shorter between runs? (There is no evidence in the text for that), (2) did solution times became shorter between runs (according to the authors answer to the first reviewer, it seems they actually got longer). Unless at least one of this conditions is satisfied, how do we know that the worms do not just randomly swim until they find the save dark patch?

We thank the reviewer for pointing out this issue. In figure 1.A, we can see that solution times became shorter between runs. In trial 1, the solution time is significantly higher than that of trial 4 (worms were transferred to a new maze). The short solution time means that learned worms can navigate to the dark chamber in a new maze. In Figure 2.D-G, we can see that routes became shorter between trial 1 and trial 4. However, we could not evaluate the exact route distance in each trial due to technical difficulties as demonstrated in our former response. The relevant descriptions are now added to the result part. 

Lines 50-51: I would rephrase these lines or completely omit them. (I think I can guess what the general meaning here is, but it is badly written and confusing and I am not sure how much it contributes to the overall understanding of the design).

We thank the reviewer for pointing out this issue. The sentence is now rephrased.

Lines 84-85: Here I would have liked to see some reference to the photosensitive resin used. I asked for this matter (why use a photosensitive resin in such an experiment design when photosensitivity is key) to be clarified during the previous revision – and I am asking for this again now.

We thank the reviewer for pointing out this issue. There are mainly two types of 3D printing techniques currently, fused deposition modeling (FDM) and the stereolithography (SLA). The objects produced by FDM are porous and water leaking, which made us not to consider this technique in making the main body of the maze. Also, the SLA 3D printing method which uses photosensitive resin as raw material behaves better in 3D-printing speed and resolution. The photosensitivity of the resin means that during 3D fabrication, the liquid resin cures after exposed to the UV light. After curing, the resin becomes stable, solid and can no longer change its physical properties under any kind of light. Therefore, the photosensitivity of the resin had no relevance to the light in our maze experiment. The reference and the illustration above are now added to the Materials and Methods part of the manuscript.

.Line 107: please add the missing space between "changed' and "and".

We thank the reviewer for pointing out this issue. The missing space is now added.

Lines 117-119: The authors state "Light intensity variance along Wall a is less than 5 Lux, which can be neglected" - Reference is certainly needed here. Also light level should be measured not only from the wall of each maze but also in points between the maze 'entry' and maze 'solution'. From figure 1D it seems quite likely that there is might be some brightness gradient there. 

We thank the reviewer for pointing out this issue. Actually, what we wanted to express is that in a typical maze, the light intensity variance (LIV) along wall b is more than 100 Lux, which is much larger than its LIV along wall a, which is less than 5 Lux. So, if the worms relied on the LIV to navigate, we think that they would rely more on the LIV along wall b than wall a. To avoid misunderstanding, we have now deleted the phrase “which can be neglected”. Based on illustration above, the light level between maze 'entry' and maze 'solution' should be ±5 Lux from our measurement in figure1.D, Based on this, we think it is no need to measure light intensity from everywhere in the maze. 

Line 123: This is the first time the term "latency" appears in the text. I do not understand what it means here (or for that matter, when you use it again and again in the following parts of the text). Please define this terms clearly as it seems important to your claims. I would have preferred for it to be discusses or defined in the materials and methods section or in experiment design – but it doesn't really matter as long as you define it clearly the first time it appears in the text.

We thank the reviewer for pointing out this issue. The relevant definition is added from line 71-75 in the introduction part.

General remarks for the "statistics" chapter:

This is poorly written and does encourage trust in your results. Please phrase your research questions (preferably at the end of the introduction part), then report the statistical tests you chose to test these questions and why. This part should also include the number of repetitions and of worms in each state (not only of the ones used in the tests, but also the worms disqualified and the reasons for doing so).

Important: Note that you are mentioning specific tests in the "statistics" section and then report the result of *different* tests in the result section.

We thank the reviewer for pointing out this issue. The research question is now added to the end of the introduction part and illustrations of statistical tests are now added to the statistics chapter.

Lines 142-143: This would be a good place to state the total number (N) of the worm which the authors actually included in the statistics. Also, I do not completely understand what "latency' mean here too – please add an explanation. While we are on this subject – please state clearly not only the N for participating worms, but also for the disqualified ones.

We thank the reviewer for pointing out this issue. The number of worms counted in each trial is now added to the statistics part. The definition of latency is added from line 71-75 in the introduction part.

Lines 145-145, 150-151: I would choose a term other than 'escape' and clearly define it. If you choose to remain with 'escape', please add a definition or an explanation for it (does it mean 'solved the light maze?', 'escaped the arena'? something else?). Also fix the extra dot/period in line 150. Note that in line 207 in the discussion (for example) you are using 'escape' in yet another context.

We thank the reviewer for pointing out this issue. The relevant definition is added from line 71-75 in the introduction part. The extra dot/period in line 150 is now fixed.

Lines 147-157:

a. these lines states: "To test whether this behavior is stable after changing the maze, we continued trial 5 and trial 6. comparing trial 4 to trial 2, 3, 5 and 6, there is no significant difference, showing that the worms might not use chemical cues in this case (One way ANOVA, P>0.1; Fig.2A)". The text does not specify what was compared to what?! What was the comparison criteria? Speed? Number of solutions? What is behind the statement that this comparison shows "that the worms might not use chemical cues in this case"

We thank the reviewer for pointing out this issue. The detailed explanation of the sentence is now added.

b. If there is significant difference between trial 1 and all other trails, indicate it - as you did in 2A. But this is a minor problem. If solving speed declines with trials (as stated in lines 154-157), how can you tell the worms learned? How can you say that there were more "solves" (if that is what 'escapes' means) in advanced trials because of learning and because of random chance (e.g., worms explored the maze for longer times).

We thank the reviewer for pointing out this issue. There is a logical relationship that if the speed declines with trials, it takes more time for the worms escape the maze, which makes it harder for worms to show their learning effects. However, the decline of the speed did not affect the worms’ escape latencies, so it won’t affect the conclusion that worms learn to escape the maze faster after training.

c. I ask again, what exactly is "latency" in your experiment system? Also, why use 241mm. in some cases and actual moving distance in others? You are pooling two different data sets together and then running a single test on the new pool? Why? Please explain this.

We thank the reviewer for pointing out this issue. The definition of latency is added from line 71-75 in the introduction part. 241 mm is the perimeter of the maze. For worms which can quickly get into the dark chamber, its moving distance is usually less 241 mm, so we can only use its actual moving distance to count its speed. For other worms who usually climb along the wall of the maze, it is more accurate to use the maze perimeter to count its speed, so we chose 241 mm to count its speed.

Lines 162-163: Please state how many worms we are talking about (this is already the results section and still we don't know how many worms were used for the statistics). What are the learning effects? As mentioned before – if solution times does not diminish and/or solution routes does not shorten, what is the criteria for learning?

We thank the reviewer for pointing out these issues. These issues can be solved by my responses on other issues above: through definition of latency and illustration of effect of worm speed on latencies. 

Line 169: Please add a space between "40" and "min.

We thank the reviewer for pointing out this issue. The space is now added.

Lines 171-174: please fix this entire paragraph: the worms' routes Illustrations starts with 2D and not with 2C as appears in the text. More important issue here is that for some reason, in the advanced trials illustrations the end point of the worms *are not* under the dark chamber (while ending under it in the first trial). It seems like, at least from the illustrations that in opposition to the authors claims – the worms did not learn.

We thank the reviewer for pointing out this issue. The worms' routes Illustrations are now fixed. The end points were outside the dark chamber is because our camera could not catch the worm routes inside the dark chamber. Anyway, all worms that solved the maze were identified that they were in the dark chamber after the test. And we can provide videos to demonstrate this issue.

General remarks for the "Results" chapter:

I would like to see a better written Results part. This should include better definitions of what was measured. (for example: what is 'latency'? what are 'escapes'? how was 'speed' calculated? Why there is no report regarding routes length). Please describe exactly what was measured or tested under each such definition or terms. Do so before or when you are introducing this term for the first time.

We thank the reviewer for pointing out these issues. These issues can be solved by my responses on other issues above: through definition of latency and illustration of effect of worm speed on latencies, and illustration of routes length. 

You should also state what exactly was compared to what? Did solution times change between trial 1 and 4? If yes - To what direction? If not – why and what criteria was compared? Why did you choose to compare trial 1 to 4 and not 1 to 6?

We thank the reviewer for pointing out these issues. These issues can be solved by my responses on other issues above: through definition of latency and illustration of effect of worm speed on latencies, and illustration of routes length.

Trial 4 uses newly manufactured mazes to test the worms, in this case, worms cannot use local cues such as chemicals for navigation, so we compare trial 1 to 4 and not 1 to 6. This illustration locates at the first sentence of the result section.

I would also like to see the actual number of animals eventually used in each statistical test as this is very unclear in the current text (something is mentioned under fig.2 but it is unclear). The N should appear also in the manuscript main text next to each relevant test. This is especially true when some of the N values mentioned are a bit too small for comfort).

Eventually, I cannot understand why you are describing statistical tests in the 'statistics' section and report results from completely different tests in the results. Please take care of this issue.

We thank the reviewer for pointing out these issues. We have now added detailed explanation of these issues in the “Statistical Analysis” chapter.

Lines 177-184: These lines belong in the introduction. Please start the discussion with shortly describing your main findings. Then you can move on to explain them in relation to other work in the field or if there is none – their novelty. Either way though - these lines are not relevant here.

We thank the reviewer for pointing out this issue. The discussion part is now started with shortly describing our main findings.

Line 192: It seems like the word 'including' is unnecessary here.

We thank the reviewer for pointing out this issue. The word including is now deleted.

Lines 200-211: these lines also do not belong in the discussion. The process of manufacturing 3d mazes, their toxicity and the material used should appear in the introduction and in the method parts. You can of course stress the novelty and advantages of your design and techniques in the discussion – but not to delve into a deep discussion of materials and method. It belongs elsewhere. Also, please notice that in line 207, you are using 'escape' in yet another context.

We thank the reviewer for pointing out this issue. Related parts are moved to the method part.

Lines 212-214: These are the lines the discussion part should have begun with. However:

a. you are using 'escaped' in yet another context here.

b. It is not clear if the results are neither significant or actually support the authors claims.

c. This brings us back to the results section. Please rewrite it.

We thank the reviewer for pointing out these issues. These issues can be solved by my responses on other issues above: through definition of latency and illustration of effect of worm speed on latencies, and illustration of routes length. 

Lines 214-218: What does this mean for the results? How many times did this happen? This should be stated in the results section. Also, what do the authors mean by "worms cannot accurately solve every task"? Did they control for each worm in successive trials and singled out ones that repeatedly could not solve their light maze? Something else? Since there is no report of the proportions of 'solvers' vs. 'non-solvers' the reader can not evaluate the meaning or strength of this statement.

We thank the reviewer for pointing out these issues. We have now added these information in the result part. However, there are only 3 worms in 36 worms that could solve all 5 trials. Other worms can only solve 0-5 trials for some unknown reasons. 

Lines 230-234: In these lines the authors claim that "In this work, planarian worms were found to use distal cues instead of proximal" and that "planarians are now the most primitive animal found to have spatial learning ability".

In light of the remarks above, I am yet to be convinced: What were the definitions for proximal and distal in this experimental setup? According to what hypothesis or tests the authors reached these results? These should appear in the manuscripts, ideally at the end of the introduction.

We thank the reviewer for pointing out these issues. However, we did not explore which kind of proximal and distal cues that the worms relied on. Anyway, we listed possible proximal and distal cues in the 3rd paragraph in the introduction part.

General remarks for the discussion:

The discussion also need some rewriting: The entire opening paragraph (lines 177-184) describes the history of the research field and belongs (if at all), in the introduction. The discussion then moves to explain reasons for the results (which should have been briefly summarized in the opening of the discussion). Directly after that the text explains the benefits (and some of the problems) of 3D printing and only on line 212 (35 lines into the discussion) finally some of the results first appear. This is structurally wrong and needs to be fixed.

We thank the reviewer for pointing out these issues. These issues can be solved by my responses on other issues above.

What worries me more, however, is that what described in the 'results' section not necessarily shows that the worms actually learned between sessions: what is the criteria? Learning speed? Total numbers of worms that found the shelter in each trial? Shortening routes? Did the researchers followed specific worms and saw better performance? The results chapter lack clarity and the research hypothesis are not mentioned, therefore it is hard to understand what on what bases the authors' states these results.

We thank the reviewer for pointing out these issues. These issues can be solved by my responses on other issues above: through definition of latency and illustration of effect of worm speed on latencies, and illustration of routes length.

---

## [Editor Report · Decision Letter 2]

21 Jun 2023

Spatial Localization Ability of Planarians Identified Through a Light Maze Paradigm

PONE-D-22-32791R2

Dear Dr. Huang,

We’re pleased to inform you that your manuscript has been judged scientifically suitable for publication and will be formally accepted for publication once it meets all outstanding technical requirements.

Kind regards,

Elias Garcia-Pelegrin, Ph.D

Academic Editor

PLOS ONE
---

## [Editor Report · Acceptance letter]

11 Jul 2023

PONE-D-22-32791R2 

Spatial localization ability of planarians identified through a light maze paradigm 

Dear Dr. Huang:

I'm pleased to inform you that your manuscript has been deemed suitable for publication in PLOS ONE. Congratulations! Your manuscript is now with our production department. 

Kind regards, 

on behalf of

Dr. Elias Garcia-Pelegrin 

Academic Editor

PLOS ONE